# Malaria treatment health seeking behaviors among international students at the University of Ghana Legon

**Mathias Lwenge**[1,2]*, **Philip Govule**[3,4], **Simon Peter Katongole**[5], **Phyllis Dako-Gyeke**[1]

**1** Department of Social and Behavioural Sciences, College of Health Sciences, School of Public Health, University of Ghana, Accra, Ghana, **2** Faculty of Health Sciences, Uganda Martyrs University, Nkozi, Uganda, **3** Department of Epidemiology and Disease Control, College of Health Sciences, School of Public Health, University of Ghana, Accra, Ghana, **4** Department of Government Studies, School of Management Studies, Uganda Management Institute, Kampala, Uganda, **5** Department of Health Policy, College of Health Sciences, School of Public Health, University of Ghana, Accra, Ghana

* mathiaslwenge@gmail.com

## Abstract

### Introduction

Appropriate management of malaria demands early health seeking behaviour upon suspicion of malaria-like symptoms. This study examined malaria treatment seeking behaviour and associated factors among international students at University of Ghana.

### Methods

The study used a cross-sectional and quantitative approach. Data collection was undertaken using a structured questionnaire administered on a random sample of 264 international students. Data obtained on malaria treatment and factors influencing treatment behaviors were analyzed using IBM, SPSS Statistics version 22. Associations between individual characteristics and Malaria treatment seeking behavior was assessed by Pearson Chi-square($X^2$) test of independence. Binary logistic regression model was built using a backwards Wald approach, with variables retained at Wald p-value <0.05.

### Results

The findings show that 35% of the respondents obtained self-prescribed antimalarial at their utmost first choice of Malaria treatment. At bivariate level, a significant relationship between Malaria health-care seeking behaviour and:- Respondents continent, $X^2(1, N = 264) = 7.936$, p = .005; Service accessibility, $X^2(1, N = 264) = 7.624$, p = .006; Wait time, $X^2(1, N = 264) = 22.514$, p <0.001; Treatment cost, $X^2(1, N = 264) = 97.160$, p <0.001; Health insurance, $X^2(1, N = 264) = 5.837$, p = 0.016, and Perceived staff attitude, $X^2(1, N = 264) = 18.557$, p < 0.001. At multivariable analysis, inappropriate malaria health seeking behaviours was associated with low perceived service accessibility as ($\geq$30mins) (aOR = 6.67; p<0.001), perceived long wait time ($\geq$30mins), (aOR = 5.94; p = 0.015), perceived treatment

**Data Availability Statement:** All relevant data has been provided to the editors upon and for the researchers who meet the criteria of access to confidential data. This is because the University of

Ghana has put legal restrictions on sharing a de-identified data set. Please contact them on thesisoffice@ug.edu.gh to permit you have access to the data.

**Funding:** The authors received no specific funding for this work.

**Competing interests:** The authors have declared that no competing interests exist.

cost affordability (<15 GHC) (aOR = 19.88; p<0.001) and age group: -34-41years (aOR = 8.83; p<0.001).

## Conclusion

There were widespread inappropriate health-care seeking behavior for Malaria treatment among international students. Improving accessibility to malaria treatment services, reducing wait time at health facilities and the treatment cost will address inappropriate malaria treatment health seeking behaviours among the international students.

## Introduction

Malaria remains a significant global public health challenge despite extensive efforts to control it [1]. About, half of the world's population is at risk of malaria. In 2021, Malaria was prevalent in 85 countries with an estimated 247 million people reported cases and 619,000 deaths [2]. The reported malaria case incidence was 59 cases per 1000 population at risk of contracting malaria against the expected 31 cases per 1000 population in 2021 [2]. Notably, the African region with at least 234 million cases continues to account for disproportionately large share of the global malaria burden. This represents approximately 95% of the global malaria cases and about 96% malaria fatalities (593,000) in 2021 [3]. Consequently, malaria became the predominant communicable disease encountered in health facilities across Sub-Saharan Africa [4].

The West African sub-region shoulders approximately half of the global malaria burden, placing more than 300 million individuals at risk of malaria infection [5]. In Ghana, malaria remains endemic, contributing to 2% of global malaria cases and 3% of global malaria-related deaths. This positions Ghana among the top 15 countries worldwide with a high burden of malaria. Notably, malaria cases account for as much as 32.5% of all outpatient cases in health-care facilities in Ghana [6].

However, substantial progress has been achieved after the implementation of a malaria control and prevention program in Ghana. There has been a notsable 32% reduction in the proportion of malaria cases (from 237 to 161 cases per 1000 of the population) and a 7% decrease in the proportion of malaria-related deaths (from 0.4 to 0.37 per 1000 of the population at risk) in 2016 compared to 2019 [7, 8]. Malaria prevention strategies are linked to vector control methods, which involve the reduction and elimination of mosquitoes through activities such as spraying and the prevention of human exposure to potential mosquito bites by utilizing insecticide-treated bed nets [9–11].

The World Health Organization emphasizes early diagnosis and swift treatment as fundamental components of effective malaria case management [8]. Consequently, in April 2012, the Global Malaria Control Program of WHO launched the "3 Ts" initiative (Test, Treat, and Track) with the goal of enhancing malaria testing, aligning drug administration with test results, and ensuring proper case reporting. This initiative prompted the revision of malaria treatment guidelines in 2010, mandating that cases be tested and treatment administered based on results from tests like RDT or microscopy [12]. These actions reflect responsible health-seeking behavior when individuals suspect they have symptoms and signs of malaria [10, 13].

Cultivating appropriate health-seeking behavior for malaria treatment is crucial for successfully preventing malaria-related mortality [11, 13]. It is important to note that malaria-like symptoms can be indicative of other febrile conditions beyond malaria [8, 10]. Therefore,

individuals experiencing fever should seek care at a suitable healthcare facility to rule out malaria or other potential fever causes [14]. Incorrect decision-making regarding malaria management can result in delayed diagnosis and treatment, thereby increasing the risk of severe disease and even death [2, 8].

Health-seeking behavior encompasses the combined actions people take to address any perceived health issues [15]. Nevertheless, certain factors, such as access to healthcare facilities, individuals' knowledge and attitudes regarding malaria, the availability of health insurance coverage, and socio-economic conditions, can impede the adoption of suitable malaria health-seeking behavior among individuals seeking care at healthcare facilities [16, 17].

Variations in malaria risk exist between different populations. Ghana poses a heightened risk of malaria to individuals from malaria-free countries, where less virulent Plasmodium species are predominant [18]. This heightened risk can be attributed to their limited immunity to malaria and lack of prior exposure to the Plasmodium parasite. International students, including those from North America and Europe, are not only susceptible to malaria and its associated complications upon arriving in Ghana, but they also encounter various barriers to accessing healthcare services. Likewise, students from regions where malaria is endemic may exhibit malaria treatment-seeking behaviors and healthcare practices that differ from those expected of in Ghana. Delay in seeking malaria treatment increases the risk of inappropriate treatment [19]. Consequently, students from non-African, non-malaria endemic countries become particularly vulnerable to severe malaria outcomes, especially during the rainy season, when mosquito breeding is more rampant than in other seasons. It is crucial for international students to adopt effective malaria prevention and healthcare-seeking behaviors while at university.

Despite an abundance of information on the prevalence and treatment behaviors of imported malaria among students returning from high-malaria-burden countries, there is limited data regarding their healthcare-seeking behaviors after contracting the disease while in regions with high malaria prevalence. This gap in knowledge pertains not only to Ghana but also to sub-Saharan Africa, highlighting the need for further research on timely malaria health-seeking behavior among international students, a significant segment of global travelers.

## Methods

### Study design, setting and population

A quantitative cross-sectional descriptive study was conducted to evaluate the behaviors of international students at the University of Ghana when seeking malaria treatment and to identify the factors associated with these behaviors. The university is equipped with two hospitals, a sick bay, and several private pharmacies conveniently accessible to students. This offers them primary healthcare services at relatively low or no cost. The research specifically targeted international students, who make up 2.14% of the university's total student population, totaling 840 students as of July 2018 [20]. The study was conducted across the four university colleges.

### Sample size and sampling procedures

The study's sample size was determined using the single proportion formula: $n = \frac{Z^2 PQ}{\delta^2}$ [21], where: n represents the minimum required sample size, Z is the standard normal deviation at a 95% confidence level (1.96), p is the estimated proportion of students exhibiting good malaria health-seeking behavior. Since the prevalence of this behavior among international students was unknown, a conventional estimate of 50% (P) was used, Q is the complement of

p (Q = 1−p) and δ is the desired level of precision (0.05) at a 95% confidence interval. After adjusting for a finite population correction and considering a 10% non-response rate, the final sample size required was 291 students. A systematic sampling approach was employed to select participants from a predefined sampling frame containing all registered international students, sourced from the university's International Program Office (IPO). We selected 840 international students from a population of 840 by randomly selecting a number, 221, and then selecting every third person on the list. A fever, headache, joint pain, general weakness, loss of appetite, and body pains were symptoms suggestive of malaria that participants must have experienced while they were at the University of Ghana. The study included only international students enrolled in 2017–2018.

## Data collection procedures and instruments

A survey was conducted at the university using a self-administered structured questionnaire between March and June 2018. Data were collected regarding participants' sociodemographic characteristics, enabling factors, knowledge about malaria, and past treatment-seeking behaviors when experiencing malaria-like symptoms. A questionnaire was administered to participants to assess their malaria knowledge, including questions on malaria transmission, mosquito breeding sites, signs and symptoms, preventive measures, and sources of information about malaria. The first author (ML) contacted each participant, explained the study's purpose, obtained consent, and provided the questionnaire for completion upon meeting the inclusion criteria and agreeing to participate. Contact information, including phone numbers and halls of residence, was obtained from the student list at the International Program Office (IPO).

## Study variables and scale of measures

The study concentrated on malaria healthcare-seeking behavior during the participants' most recent experience of malaria-like symptoms while at the university. The primary variable of interest was whether international students at the University of Ghana sought treatment for suspected malaria at a health facility or elsewhere. To assess this variable, participants were asked the following question: "During your last illness, where did you seek treatment when you experienced signs and symptoms suggesting malaria?" The response to the question was transformed into a binary variable whereby those who sought malaria treatment at a 'health facility' (as defined earlier) were categorized as 'Yes = 1.' Conversely, individuals who sought care from sources other than a health facility (such as traditional healers, herbalists, drug shops without a proper diagnosis, or those who did not seek any treatment despite suspecting malaria) were categorized as 'No = 0' for their last suspected malaria episode, indicating that they sought treatment outside of a healthcare facility. This characterization is consistent with a study conducted in southeastern Tanzania [22].

Data were collected about several independent variables, guided by the existing literature on the behavioral model [23]. This Health belief model helps to understand the health-seeking behaviours for malaria treatment among international students. It emphasizes a need to express a lot of care through empathy and clear understanding of the student's sick role during malaria illness. This is through the proper use of evidence-based health information, education and communication, (IEC) which can help to improve on the health-seeking behaviours for malaria treatment The primary categorical independent variables of interest include: (i) Service Accessibility: Categorized as more accessible (if it takes less than 30 minutes to reach the facility) or less accessible (if it takes ≥30 minutes). (ii) Waiting Time: Classified as short (<30 minutes) or long (≥30 minutes). (iii) Affordability: Grouped as more affordable (if the cost

was ≥15GHC) or less affordable (if it required paying <15GHC). (iv) Cost of Transport to seek care: Categorized as more affordable (if the cost was <8GHC) or less affordable (if it was ≥8GHC). (v) Health Insurance: Indicated as "No" or "Yes." (vi) Perceived Service Availability: Indicated as "No" or "Yes." (vii) Knowledge of Malaria: Distinguished as "Knowledgeable" or "Not knowledgeable." (viii) Perception of Health Worker Attitude: Categorized as "Friendly" or "Not Friendly."

Additional factors of interest include: (ix) Residential Status: Whether students reside at the university (on campus) or off-campus (outside the university premises). (x) Sponsorship Status: Whether students were self-sponsored or on a paid scholarship. (xi) Health Insurance: Indicated as "Yes" or "No." Furthermore, socio-demographic characteristics were controlled and these included: (xii) Age: Categorized as ≤29 years, 30–39 years, 40–49 years, or >50 years. (xiii) Sex: Male or female. (xiv) Marital Status: Married or single. (xv) Level of Education: Bachelor's, master's, or doctorate. (xvi) The continent of Origin: Africa or other continents. (xvii) College of Study: Health Sciences, Basic and Applied Health Sciences, Humanities, and College of Education. Continuous variables, such as age, were summarized with a mean and standard deviation.

## Data management and analysis

The data underwent completeness and consistency checks before it was coded and entered into a pre-designed Excel template, which was subsequently exported to SPSS statistical software version 22 (IBM, V22) for analysis. Descriptive statistics were employed to organize and summarize the data, presenting frequencies and proportions for categorical variables and mean values for continuous variables at the univariate level. A bivariate analysis was conducted using Pearson's Chi-square statistics ($X^2$) to examine the relationship between categorical variables and the seeking of malaria treatment from healthcare facilities. This was possible as all variables satisfied the Pearson $X^2$ assumptions, such as having cell counts above five.

A multivariate binary logistic regression analyses were conducted to assess the impact of predictors or explanatory variables on the outcome variable. In the multivariable model, it was assessed alongside other variables. A backward Wald approach was used to build multivariable logistic regression models. Explanatory variables associated with the outcome of interest (malaria treatment-seeking behavior) at a significance level of 0.2 (p≤0.20) were included in the multivariable regression. Predictor variables with a Wald p-value <0.05 were retained in the final multivariable models, signifying their independent association with the choice of malaria treatment-seeking behavior. The results were expressed as adjusted odds ratios (aOR) with a 95% confidence interval (CI). The final model's goodness of fit was assessed using Hosmer and Lemeshow's test. Before performing multivariable logistic regression, a multicollinearity test was also executed via multiple linear regression, excluding variables with Variance Inflation Factors (VIF) greater than 10. The results indicated no evidence of multicollinearity among the explanatory factors.

## Ethics approval, procedures and consent to participate in the study

Ethical clearance to conduct the study (Approval No: GHS-ERC023/12/17) was sought from and granted by the Ghana Health Service Ethics Review Committee (GHS-ERC). Permission was also granted by the Dean, International program office (IPO)—University of Ghana, the dean of students and Senior Hall Tutor for Jubilee Hall and the International Students' Hostel where most international students reside. All other ethical procedures were in the same way followed including the fact that written informed consent was obtained from participants before including them in the study. To ensure confidentiality, initials and questionnaire

number codes were used instead of real names, hence no personal identifiers included in at analysis level of the study.

## Results

### General characteristics of the respondents

Slightly over half of the participants (51.1%, 135 out of 264) were male. The respondents' ages ranged from 18 to 39 years, with an average age of 25.2 years. More than half of the participants (58.7%, 155 out of 264) were in the 18–25 age bracket. The majority of respondents (71.2%, 188 out of 264) hail from African countries. Around 54.2% of respondents possess valid health insurance coverage. A significant portion (87.1%, 230 out of 264) resided either on the main university campus or on the Korle-Bu medical school campus. A majority, 204 out of 264 (77.4%), identified themselves as Christians and were either unmarried or single (79.5%, 210 out of 264). The College of Basic and Applied Health Sciences had the highest representation (39.4%) of respondents, while the College of Education had the lowest (8%). Table 1 reveals the general characteristics of the respondents.

### Malaria treatment health-seeking behaviors

The investigation into respondents' initial actions upon experiencing malaria-like symptoms revealed several treatment-seeking behaviors. Orthodox care options included purchasing medicines from a pharmacy, either solely for anti-malaria treatment or for both antimalarials and analgesics. Non-orthodox behaviors encompass seeking care at a health facility or resorting to alternative treatments like herbs, prayers, or warm baths. A substantial majority (86.3%) chose to buy drugs from a pharmacy as their first malaria treatment option. Among them, 35.2% purchased anti-malarial drugs alone, 26.9% bought both antimalarials and analgesics and 24.2% purchased only analgesics. A mere 2.7% of the 264 respondents went directly to a health facility as their first point of care for suspected malaria. A slight disparity emerged in the proportion of males and females opting for pharmacy-based treatment (43.9% for males and 42.5% for females).

**Table 1. Actions taken to seek for health care by sex and their continent of residence (N = 264).**

| Action | Total (N = 264) | Sex | | Continent | |
| --- | --- | --- | --- | --- | --- |
| | | Males | Females | Africa (n = 188) | Non-African (n = 76) |
| | N (%) | n(%) | n(%) | n(%) | n(1%) |
| **The first line of action** | | | | | |
| Antimalarial only from a pharmacy | **93(35.2)** | 53(39.3) | 40(31%) | 61(32.4) | 32(42.1) |
| Antimalarial +painkillers pharmacy | **71(26.9)** | 38(28.1) | 33(25.6) | 60(31.9) | 11(15.4) |
| Painkillers only from a pharmacy | **64(24.2)** | 25(18.5) | 39(30.2) | 41(21.8) | 23(30.3) |
| Went to a health facility (hospital/clinic | **10(2.7)** | 7(3.7) | 3(1.6) | 10(3.7) | 0(0.00) |
| Non-medical actions | **26(11)** | 12(10.4) | 14(11.6) | 16(10.2) | 10(12.2) |
| **The second line of action** | | | | | |
| Health facility (hospital/clinic) [a] | **152(57.6)** | 79(58.5) | 73(56.6) | 98(52.1) | 54(71.1) |
| Pharmacy (drug shop) | **101(38.3)** | 53(39.3) | 48(37.2) | 80(42.6) | 21(27.6) |
| Non-medical actions [b] | **11(4.2)** | 3(2.2) | 8(6.2) | 10(5.3) | 1(4.2) |
| **Total** | **264(100)** | **135(100)** | **129(100)** | **188(100)** | **76(100)** |

a = Included those who had first sought for care at health facility and returned for review

b = Herbs/ prayers/warm bath

Across African and non-African students, there was no significant difference in the proportion seeking treatment from a pharmacy initially (86.1% for Africans and 87.8% for non-Africans). Among non-African students, a substantial portion (42.1%) bought antimalarials as their first treatment. Among African international students, the usage of either antimalarials alone (32.4%) or both antimalarials and analgesics (31.9%) from a pharmacy was equally common. Notably, only 3.7% of African international students sought treatment from a health facility initially, while no non-African international students did so.

Regarding subsequent actions when initial treatment did not lead to improvement, 57.6% of respondents turned to health facilities, while 38.3% sought further care from pharmacies or drug shops. Male and female respondents showed a minor discrepancy in seeking care from health facilities (58.5% for males and 56.6% for females). A greater proportion of non-African international students (71.1% of 76) sought care from health facilities as the second option compared to African international students (52.1%).

## The relationship between participants characteristics and malaria treatment seeking behaviour at bivariate analysis

The relationships between malaria treatment-seeking behavior and participant characteristics are summarized in Table 2. Firstly, a significant association emerged between malaria treatment healthcare-seeking behavior and the continent of the respondent: $X^2$ (1, N = 264) = 7.936, p = .005. Participants from continents other than Africa were more likely to appropriately seek Malaria treatment for suspected febrile illnesses compared to those from Africa. Moreover, a significant correlation was observed between perceived service accessibility and Malaria treatment-seeking behavior, $X^2$ (1, N = 264) = 7.624, p = .006. Students who perceived services as less accessible, requiring at least 30 minutes for access ($\geq$30mins), were more inclined to seek appropriate Malaria treatment. Similarly, malaria treatment-seeking behavior demonstrated a statistically significant connection with perceived waiting time, $X^2$ (1, N = 264) = 22.514, p < 0.001. Those who perceived longer waiting times ($\geq$30mins) were more likely to adhere to recommended malaria treatment-seeking behavior by seeking care at health facilities.

Furthermore, a substantial association between malaria treatment-seeking behavior and perceived treatment cost was evident. $X^2$ (1, N = 264) = 97.160, p < 0.001. Respondents who considered treatment costs affordable (<15 GHC) were more likely to seek appropriate malaria treatment from health facilities. A significant relationship also existed between malaria treatment-seeking behavior and having health insurance coverage; $X^2$ (1, N = 264) = 5.837, p = 0.016. Participants without health insurance were more likely to adhere to recommended malaria treatment-seeking behaviors by seeking care at health facilities. Finally, a significant association was identified with perceived staff attitudes, $X^2$ (1, N = 264) = 18.557, p < 0.001. Respondents who perceived staff attitudes as friendly were more inclined to seek appropriate malaria treatment.

However, no significant association emerged between malaria treatment-seeking behavior and participant variables, including age groups (p = 0.115), sex (p = 0.751), residence status (p = 0.831), religion (p = 0.785), marital status (p = 0.116), college of respondent (p = 0.419), cost of transport (p = 0.507), service availability (p = 0.150), and knowledge of malaria (p = 0.497).

## Predictors of malaria treatment (healthcare) seeking behaviour

After adjusting for confounders and accounting for socio-demographic variables in the multivariable model, four explanatory variables remained significantly associated with appropriate

**Table 2. Characteristics of respondent in the study and relationship with malaria treatment behaviour (N = 264).**

| Variable | Total (N = 264) | Malaria Treatment/Health-care Seeking Behaviour (HSB) | | Bivariable | Multivariable P value |
| | | Sought care at health facility, N (%) | Did not seek care at health facility N (%) | X² | P-value aOR(95% CI) |
|---|---|---|---|---|---|
| **Sex** | | | | | |
| Male | 135 | 79(58.5) | 56(41.5) | 0.101 | 0.751 |
| Female | 129 | 73(56.6) | 56(43.4) | | |
| **Age category** | | | | | |
| 18–25 | 155 | 82(52.9) | 73(47.1) | 4.331 | 0.115   1 |
| 26–33 | 96 | 60(62.5) | 36(37.5) | | 2.50 (1.07–5.48) **0.032** |
| 34–41 | 13 | 10(76.9) | 3(23.1) | | 8.83 (1.26–61.94) **0.028** |
| **Continent** | | | | | |
| Africa | 188 | 98 (52.1) | 90(47.9) | 7.936 | 0.771 |
| Non-Africa | 76 | 54(71.1) | 22(28.9) | | |
| **Residential status** | | | | | |
| Off campus | 34 | 19(55.9) | 15(44.1) | 0.046 | 0.831 |
| On campus | 230 | 133(57.8) | 122(42.2) | | |
| **Religion** | | | | | |
| Christian | 204 | 120(58.8) | 84(41.2) | 0.818 | 0.785 |
| Muslim | 45 | 25(55.6) | 20(44.4) | | |
| Traditionalist/Atheist | 15 | 8(46.7) | 7(53.3) | | |
| **Marital status** | | | | | |
| Single | 210 | 126(60) | 84(40) | 2.47 | 0.116   1 |
| Married | 54 | 26(48.1) | 28(51.9) | | 0.36 (0.12–1.05) 0.60 |
| **College** | | | | | |
| Health Sciences | 61 | 34(55.7) | 27(44.3) | 2.83 | 0.419 |
| Humanities | 78 | 29(37.2) | 49(62.8) | | |
| Basic & Allied Science | 104 | 60(57.7) | 44(42.3) | | |
| Education | 21 | 9(42.9) | 12(57.1) | | |
| **Perceived service accessibility** | | | | | |
| More accessible<30mins | 227 | 123(54.2) | 104(45.8) | 7.62 | **0.006**   1 |
| Less accessible ≥30mins | 37 | 29 (78.4) | 8(21.6) | | 6.67(2.35–18.93) **<0.001** |
| **Perceived Waiting time** | | | | | |
| Short (<30mins) | 27 | 4(17.9) | 23(82.1) | 22.51 | **<0.001**   1 |
| Long (≥30mins) | 237 | 148(62.4) | 89(37.6) | | 5.94(2.35–18.93) **0.015** |
| **Treatment Cost** | | | | | |
| More affordable (<15 GHC) | 131 | 115(87.8) | 16(12.2) | 97.16 | **<0.001**   19.88(9.77–40.46) **<0.001** |
| Less affordable≥15GHC | 133 | 37(27.8) | 96(72.2) | | 1 |
| **Cost of transport** | | | | | |
| More Affordable <8GHC | 114 | 63(55.3) | 51(44.7) | 0.439 | 0.507 |
| Less affordable ≥ 8GHC | 150 | 89(59.3) | 61(40.7) | | |
| **Health Insurance** | | | | | |
| No | 143 | 92(64.3) | 51(35.7) | 5.837 | 0.871 |
| Yes | 121 | 60(49.6) | 61(50.4) | | |
| **Perceived service Availability** | | | | | |
| No | 198 | 109(55.1) | 89(44.9) | 2.688 | 0.150 |
| Yes | 66 | 43(65.2) | 23(34.8) | | |

*(Continued)*

**Table 2.** (Continued)

| Variable | Total (N = 264) | Malaria Treatment/Health-care Seeking Behaviour (HSB) | | Bivariable Multivariable P value | |
| | | Sought care at health facility, N (%) | Did not seek care at health facility N (%) | $X^2$ | P-value aOR(95% CI) |
| --- | --- | --- | --- | --- | --- |
| **Malaria Knowledge** | | | | | |
| Knowledgeable | 229 | 130(56.8) | 99(43.2) | 0.461 | 0.497 |
| Inadequate knowledge | 35 | 22(62.9) | 13(37.1) | | |

[5]Odds ratios adjusted for age group, sex and marital status

[6]Hosmer and Lemeshow test of goodness of fit for the model was significant at $X^2$ = 11.283, df (8) p.value = 0.186; Nagelkarke Pseudo $R^2$ = 0.553; Cox & Snell $R^2$ = 0.412; log likelihood = 219.788; GHC Ghana Cedis

malaria treatment-seeking behavior. These variables were: perceived service accessibility (p < 0.001), perceived waiting time (p = 0.015), treatment cost (p < 0.001), and respondent age group. Perceived service accessibility remained significant, indicating that respondents perceiving services as less accessible ($\geq$30mins) were approximately 7 times more likely to adhere to appropriate malaria treatment-seeking behavior compared to those perceiving services as more accessible (<30mins) (aOR = 6.67; 95% CI: 2.35–18.93, p < 0.001). Likewise, perceived waiting time remained a strong predictor, with respondents experiencing longer wait times ($\geq$30mins) being about 6 times more likely to seek appropriate health care (aOR = 5.94; 95% CI: 2.35–18.93, p < 0.015). Participants finding treatment costs more affordable (<15 GHC) exhibited a significant tendency to seek appropriate health care (aOR = 19.88; 95% CI: 9.77–40.46, p < 0.001). Additionally, age was significant, with those aged 34–41 years being around 9 times more likely to seek appropriate health care compared to the younger age group (18–25 years) (aOR = 8.83; 95% CI: 1.26–61.94, p < 0.028), and those aged 26–33 years being about 2.5 times more likely (aOR = 2.50; 95% CI: 1.07–5.48).

## Discussion

This study delved into the actions taken and factors influencing the health-seeking behaviors of international students at the University of Ghana in relation to malaria. This is because malaria treatment plays a crucial role in the malaria control program as an essential intervention to mitigate the burden of this parasitic disease, which claims numerous lives [24]. The investigation followed an approach that categorizes health-seeking behavior into "first" and "second" lines of action taken by individuals who feel unwell after experiencing symptoms resembling malaria.

In the first line of action, a substantial proportion (35.2%) of international students resorted to self-prescribed antimalarial drugs without adherence to the test and treat plus trace policy (3Ts). These findings correspond a study among university students in South Western Nigeria, which revealed that 46.3% of them used antimalarial drugs for self-treatment upon experiencing malaria-like symptoms [25]. Similarly, the outcomes of this study align with another research conducted among Nigerian university students, where 66% utilized various medicines, including antimalarials, to address malaria-related symptoms [26]. Notably, a significant number (65.6%) of international students from African countries opted for this line of action when experiencing malaria-related symptoms. Previous studies have indicated that self-medication is common among African patients for malaria [9, 32]. It's important to note that inappropriate use of antimalarial medications as seen in this study can contribute to antimicrobial resistance, particularly when drugs are taken without confirmed presence of parasites in the body [27]. Therefore, more action is required to end this practice among UG's international students.

In terms of the second line of action, a majority (57.6%) of respondents sought care at health facilities such as hospitals or clinics for further treatment of malaria-related symptoms, compared to 38.3% who sought care from pharmacies and other non-formal sources. These findings contrast with a study in Nigeria, where only 24.7% of university students sought care from health facilities, with a preference for self-medication by purchasing drugs from pharmacies [28]. This increased utilization of health facilities among international students at the University of Ghana can be attributed to the easy accessibility of facilities like Legon Hospital and the university clinic. These facilities are conveniently located on the university campus, making it convenient for students to seek prompt medical care when experiencing malaria symptoms. This shows an impressive achievement compared to Nigeria's low formal health facility utilization rate of international students at just 12% [26].

In this study, predictive factors for malaria health-seeking behavior among 264 international students were explored. Respondents from non-African countries (42.1%) were more likely to seek services from health facilities as a second line of action during their most recent malaria treatment episode. Accessibility, the perception of waiting times, the cost of treatments, and health insurance coverage all influence this behavior. The Andersen-Newman Framework for Health Services Utilization elucidates those various factors, including individual-level characteristics, influence the uptake of health services [29]. Health insurance was associated with a higher likelihood of seeking care from health facilities, possibly because insured individuals do not need to pay out-of-pocket fees at the time of care-seeking. This finding is consistent with previous studies showing the impact of health insurance on health-seeking behavior, indicating that it reduces the financial barriers to accessing care [30, 31]. The finding underscores the importance of international students having health insurance to facilitate easy access to treatment while ill.

Regarding accessibility to health services, a significant proportion (46.6%) of international students had easy access to health facilities, contributing to a higher number seeking care from formal health institutions. This is attributed to the presence of the university clinic and Legon Hospital in close proximity to where most international students reside. Students can easily reach these facilities in 30 minutes or less. This easy accessibility was found to increase the likelihood of seeking care from health facilities, making it 6.4 times higher when compared to distances of over 30 minutes away from their residence halls. This trend is consistent with studies conducted in Nigeria and Ethiopia, where access to health facilities among students was reported to influence health-seeking behavior [32, 33]. The findings inform the university administrators of the need to maintain health facilities that students can easily use as the university grows in the future. Similarly, to this, prospective international students should be counseled to always choose accommodation close to medical services to ensure good care when ill.

The perception of waiting time also significantly played a role in malaria treatment health-seeking behavior. Respondents who had to wait for 30 minutes or more at a facility were about 6 times more likely to seek appropriate health care for malaria treatment compared to those perceiving shorter wait times. This can be attributed to the fact that those who sought care at health facilities underwent diagnostic procedures and treatment, which naturally took longer than non-formal sources of care. International students should be encouraged not to be discouraged by the somewhat lengthy wait between arrival at the facility and departure because this is due to the need to provide appropriate care. This includes taking a patient's medical history and conducting investigations, which account for the somewhat prolonged waiting time but necessary for proper malaria like symptoms management.

The cost of treatment had an inverse relationship with health-seeking behavior. This was due to the provision of free medical services at the university clinic and Legon Hospital, along

with comprehensive health insurance coverage. This aligns with studies showing increased utilization of services when antimalarial drug prices are subsidized [22]. However, findings in this study differ from those in Tanzania, where factors such as poor patient-provider relationships, the inaccessibility of medicines, and associated costs deter individuals from seeking care from health facilities [34, 35]. The finding further highlights the need to have proper planning for health costs by mainly letting in students with insurance coverage. With a decent student health insurance plan, international students have the confidence that if they fall ill, they will not have to undergo costly medical care, which may deter them from having appropriate treatment-seeking behaviors for any illness.

The availability of health facilities, clinics, and pharmacies around the university campus, coupled with the proximity of these facilities to student residences, contributed to increased health facility utilization. This finding is consistent with other studies that highlight availability as a key factor in influencing health-seeking behavior [34]. Factors like service availability, transport costs, and health worker attitudes, although recognized as influencing health-seeking behavior, did not show significant associations in this study.

## Conclusion

This study looked at how international students at the University of Ghana seek treatment for malaria and what factors predict such behavior upon suspicion of malaria-related symptoms. Antimalarial self-medications were mainly used as a first line of defense. The use of a medical facility (hospital or clinic) for additional management was the second-most common course of action upon failure of the first option. Service accessibility, wait times, treatment costs, and respondents' age are four explanatory variables that significantly influence international students' malaria treatment-seeking behavior at the University of Ghana. In order to alter improper malaria treatment-seeking behaviors among these students, it is recommended to use multifaceted approaches to tackle these factors.

## Supporting information

**S1 Dataset.**
(ZIP)

## Acknowledgments

The authors acknowledge the contribution of the international students who participated in the study, and various institutions and individuals who supported the project. The views expressed in the article solely belong to the authors and do not necessarily reflect the views of the, IPO or their respective institutions.

## Author Contributions

**Conceptualization:** Mathias Lwenge.

**Formal analysis:** Mathias Lwenge, Philip Govule.

**Supervision:** Phyllis Dako-Gyeke.

**Writing – original draft:** Mathias Lwenge, Philip Govule, Simon Peter Katongole.

**Writing – review & editing:** Simon Peter Katongole, Phyllis Dako-Gyeke.

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
