## [Decision Letter · Decision Letter 0]

11 Nov 2022

PONE-D-22-27540Malaria treatment health-seeking behaviors among international students at the University of Ghana Legon, AccraPLOS ONE

Dear Dr. Lwenge,

Thank you for submitting your manuscript to PLOS ONE. After careful consideration, we feel that it has merit but does not fully meet PLOS ONE’s publication criteria as it currently stands. Therefore, we invite you to submit a revised version of the manuscript that addresses the points raised during the review process.

The authors should address all the comments suggested by the reviewers.  

We look forward to receiving your revised manuscript.

Kind regards,

Sandra Boatemaa Kushitor, Ph.D.

Academic Editor

PLOS ONE

Journal Requirements:

"Mathias lwenge (ML)

Simon Peter Katongole (SPK)

Philip Govule (PG)

Phyllis Dako Gyeke (PDG)

ML and PD-G initiated the project. ML conducted the data analysis with considerable input from PG. ML was involved in development of first draft of the manuscript with PG, SPK, and reviewed by PD-G. All authors contributed to the subsequent versions of the manuscript"

5. We note you have included a table to which you do not refer in the text of your manuscript. Please ensure that you refer to Tables 2 and 3 in your text; if accepted, production will need this reference to link the reader to the Table.

**Additional Editor Comments:**

Dear Authors,

This manuscript makes a significant contribution to the malaria treatment. However, the major flaw of the study is the lack of reference to the Ghana Health Service Standard Treatment Guidelines for Malaria. I suggest that the authors should read this document and use the recommendations of the GHS in defining their outcome variables. Additional comments can be found in the attached document.

Reviewers' comments:

Reviewer's Responses to Questions

**Comments to the Author**

1. Is the manuscript technically sound, and do the data support the conclusions?

Reviewer #1: No

Reviewer #2: Yes

2. Has the statistical analysis been performed appropriately and rigorously? 

Reviewer #1: No

Reviewer #2: Yes

3. Have the authors made all data underlying the findings in their manuscript fully available?

Reviewer #1: Yes

Reviewer #2: Yes

4. Is the manuscript presented in an intelligible fashion and written in standard English?

Reviewer #1: No

Reviewer #2: No

5. Review Comments to the Author

Reviewer #1: It is an interesting work n the field of infectious disease, but needs a major revision to be sound for the scientific community.

Reviewer #2: General comments:

1. What is the main significance of this study?

2. What new finding/insights could this study add/bring to what is already known by scientific community?

Specific comments:

1. In your abstract result part, the word “according to the findings” seems others work/study done somewhere by other researchers?

2. Font size used was not consistent/uniform, there were differences in some paragraphs?

3. How did you measure wither or not the students were knowledgeable or have inadequate knowledge regarding Malaria treatment health-seeking behaviors?

4. In conclusion part, the word “widespread” tends to generalize things only based on a single study with limited geographical coverage; better to use appropriate word?

6. PLOS authors have the option to publish the peer review history of their article (what does this mean?). If published, this will include your full peer review and any attached files.

Reviewer #1: No

Reviewer #2: No

---

## [Author Response · Author response to Decision Letter 0]

6 Jul 2023

Dear Reviewers , this feedback was taken in good faith and i have tried my level best to address the comments with the support of co-authors. Thanks so much for the good job that you accorded me to have a well written manuscript.

---

## [Decision Letter · Decision Letter 1]

21 Aug 2023

PONE-D-22-27540R1Malaria treatment -seeking behaviors among international students at the University of Ghana Legon, AccraPLOS ONE

Dear Dr. Lwenge,

Thank you for submitting your manuscript to PLOS ONE. After careful consideration, we feel that it has merit but does not fully meet PLOS ONE’s publication criteria as it currently stands. Therefore, we invite you to submit a revised version of the manuscript that addresses the points raised during the review process.

We look forward to receiving your revised manuscript.

Kind regards,

Sandra Boatemaa Kushitor, Ph.D.

Academic Editor

PLOS ONE

Journal Requirements:

Additional Editor Comments:

Dear Authors,

Reviewers have completed the review of your manuscript. Kindly edit the grammar of your work. I recommend that you use a language service agency. Also you will need to address your tables and do not repeat results in the results section.

Best regards

Reviewers' comments:

Reviewer's Responses to Questions

**Comments to the Author**

1. If the authors have adequately addressed your comments raised in a previous round of review and you feel that this manuscript is now acceptable for publication, you may indicate that here to bypass the “Comments to the Author” section, enter your conflict of interest statement in the “Confidential to Editor” section, and submit your "Accept" recommendation.

Reviewer #1: All comments have been addressed

Reviewer #2: (No Response)

2. Is the manuscript technically sound, and do the data support the conclusions?

Reviewer #1: Yes

Reviewer #2: Yes

3. Has the statistical analysis been performed appropriately and rigorously? 

Reviewer #1: Yes

Reviewer #2: Yes

4. Have the authors made all data underlying the findings in their manuscript fully available?

Reviewer #1: Yes

Reviewer #2: No

5. Is the manuscript presented in an intelligible fashion and written in standard English?

Reviewer #1: Yes

Reviewer #2: No

6. Review Comments to the Author

Reviewer #1: 1. Comments….Unless otherwise you will discuss the each results you don’t need to describe it separately…..the bivariable results..merge it with multivariable table and finish it with two –three sentence

Outcome measurement is putted for your dependent variable but not summarized in result part…simply putted as descriptive stastical your first objective ..needs to have overall result based on your operational definition.

Not corrected – still there is separate table also there is wrong term in table 3 which is Univariable

2. Conclusion- not in-line with the findings---still not corrected

Reviewer #2: 1. I note an improvement in the revised version of the manuscript.

2. Still, there are numerous typographical errors that need to be updated or fixed.

3. Issues concerning grammar use should be corrected.

4. The language ought to get better as well.

7. PLOS authors have the option to publish the peer review history of their article (what does this mean?). If published, this will include your full peer review and any attached files.

Reviewer #1: No

Reviewer #2: No

---

## [Author Response · Author response to Decision Letter 1]

4 Oct 2023

The methodology section has been revised to eliminate the univariate description on page 6 of the manuscript. 

The multivaribles have been merged in table 2 on page 8 and 9 of the manuscript.

Outcome measurement has been indicated as per the operational definition on page 7 of the manuscript 

Table 3 has been removed and merged the table with table two 

Conclusion has been revised and now in line with the study findings

Numerous typos and grammatical errors have corrected through out the manuscript

---

## [Editor Report · Decision Letter 2]

11 Oct 2023

Malaria treatment -seeking behaviors among international students at the University of Ghana Legon, Accra

PONE-D-22-27540R2

Dear Dr. Lwenge,

We’re pleased to inform you that your manuscript has been judged scientifically suitable for publication and will be formally accepted for publication once it meets all outstanding technical requirements.

Kind regards,

Sandra Boatemaa Kushitor, Ph.D.

Academic Editor

PLOS ONE
---

## [Editor Report · Acceptance letter]

17 Oct 2023

PONE-D-22-27540R2 

Malaria Treatment Health Seeking Behaviors among International Students at the University of Ghana Legon 

Dear Dr. Lwenge:

I'm pleased to inform you that your manuscript has been deemed suitable for publication in PLOS ONE. Congratulations! Your manuscript is now with our production department. 

Kind regards, 

on behalf of

Dr. Sandra Boatemaa Kushitor 

Academic Editor

PLOS ONE